# Identification of 15 T Cell Restricted Genes Evaluates T Cell Infiltration of Human Healthy Tissues and Cancers and Shows Prognostic and Predictive Potential

**DOI:** 10.3390/ijms20205242

**Published:** 2019-10-22

**Authors:** Luigi Cari, Francesca De Rosa, Maria Grazia Petrillo, Graziella Migliorati, Giuseppe Nocentini, Carlo Riccardi

**Affiliations:** 1Section of Pharmacology, Department of Medicine, University of Perugia, 06129 Perugia, Italy; luigi.cari@hotmail.it (L.C.); francescaderosa94@libero.it (F.D.R.); graziella.migliorati@unipg.it (G.M.); carlo.riccardi@unipg.it (C.R.); 2Signal Transduction Laboratory, NIEHS, NIH, Department of Health and Human Services, Durham, NC 27709, USA; maria.petrillo@nih.gov

**Keywords:** tissue-resident memory T cells, T cell infiltration of human cancer, microarray, bioinformatics, precision medicine

## Abstract

T cell gene signatures are used to evaluate T cell infiltration of non-lymphoid tissues and cancers in both experimental and clinical settings. However, some genes included in the available T cell signatures are not T cell-restricted. Herein, we propose a new human T cell signature that has been developed via a six-step procedure and comprises 15 T cell restricted genes. We demonstrate the new T cell signature, named signature-H, that differs from other gene signatures since it shows higher sensitivity and better predictivity in the evaluation of T cell infiltration in healthy tissues as well as 32 cancers. Further, results from signature-H are highly concordant with the immunohistochemistry methods currently used for assessing the prognosis of neuroblastoma, as demonstrated by the Kaplan–Meier curves of patients ranked by tumor T cell infiltration. Moreover, T cell infiltration levels calculated using signature-H correlate with the risk groups determined by the staging of the neuroblastoma. Finally, multiparametric analysis of tumor-infiltrating T cells based on signature-H let us favorably predict the response of melanoma to the anti-PD-1 antibody nivolumab. These findings suggest that signature-H evaluates T cell infiltration levels of tissues and may be used as a prognostic tool in the precision medicine perspective after appropriate clinical validation.

## 1. Introduction

T lymphocytes are present in high numbers in primary and secondary lymphoid organs and in low numbers in non-lymphoid tissues (tissue-resident memory T cells), where they are crucial for tissue homeostasis [1,2,3,4]. In some diseases, tissue-resident memory T cells of the microenvironment expand, and T cells are recruited from lymphoid organs so that T cells are found at higher levels in the diseased tissues [3,5,6]. Tumor-infiltrating T cells are the most relevant population of tumor-infiltrating lymphocytes and are found in most solid tumors (e.g., breast and colon cancer, melanoma, and neuroblastoma), where they play a crucial role in either favoring (by regulatory T cells) or inhibiting (by CD8^+^ T cells and CD4^+^ conventional T cells) tumor development [7,8,9,10,11]. In the case of a few tumors (e.g., neuroblastoma), T cell mediated tumor rejection may be observed without any treatment [12], and in many other tumors potentiation of the immune response, by conventional antitumor treatment, is a component of the efficacy of treatment [13]. Indeed, the level of T cell infiltration (Tci) is considered to be essential for the diagnosis and prognosis of many solid tumors, including neuroblastoma and colon cancer [7,8,14,15]. With the emergence of immune checkpoint inhibitors, the level of Tci of cancers may be one of the parameters used to determine the success of cancer treatment [14,16,17,18]. Indeed, tumor types that are more responsive to immune checkpoint inhibitors (e.g., melanoma and lung cancers) show massive infiltration by T cells (the so-called “hot” tumors) [7,15,19,20,21].

Tci is typically examined through immunohistochemistry methods. However, these techniques are time-consuming, and their results are affected by inter-laboratory differences. Alternatively, Tci of non-lymphoid tissues can be estimated by using T cell signatures evaluating the expression of genes that are specific for T cells, sometimes adjusting gene weight by software [22,23,24,25].

To study the Tci of cancers by bioinformatic approach (Cari et al. [26] and unpublished data), we used the hierarchical clustering tool of the Genevestigator suite to investigate, among the published T cell signatures, which one is more appropriate to analyze Tci of non-lymphoid tissues. The expression of genes included in the previously published signatures was evaluated in non-lymphoid tissues, and it was compared to the expression of the same genes in untouched and activated T cells purified from different sources. According to the results (Figure 1 and Appendix A), some genes included in the signatures were expressed at similar levels by T cells and some/several non-lymphoid tissues.

In this study, we have tested all the genes included in the previous signatures to identify a new T cell signature that includes 15 genes which are expressed at much higher levels by T cells than by non-lymphoid cells and tissues. We compared the newly identified T cell signature with the other reported signatures and assessed the signature as a tool from the precision medicine perspective, as proof-of-principle of its usefulness.

## 2. Results

### 2.1. Selection of Genes for a New T Cell Signature (Signature-H)

The objective of the study was to select T cell restricted genes so that the new T cell signature is used without the need for dedicated software, weighting the expression of the genes. To this aim, we tested the expression of all the genes (*n* = 1507) included in published T cell and T cell subset signatures [22,23,24,25,27,28,29]. In particular, expression levels of these genes by purified human T cells were used as a reference and compared with the level of the expression by purified human B cells and non-lymphoid immune cells, human cell lines, and cells from healthy tissues. We used the Genevestigator V3 suite absolute values of gene expression (log2 value) that have been generated using the Affymetrix Human Genome U133 Plus 2.0 platform were downloaded [30]. Gene expression data were obtained from datasets that are publicly available from Gene Expression Omnibus [31] and the European Bioinformatics Institute [32]. The complete list of the genes evaluated is shown in Appendix A.

In the hypothesis that the more the genes are T cell specific, the better a T cell signature performs, we selected the genes expressed at a considerably higher level in T cells than non-lymphoid cells/tissues via a six-round analysis. To establish the mean level of expression of the gene by T cells, all the available human T cells and T cell subsets were considered, including resting, memory, and activated T cells isolated from blood and lymphoid tissues.

Through rounds 1 and 2, we excluded genes that were overexpressed by less than 3.32 log2 (corresponding to ten-fold overexpression) in T cells (mean expression level) as compared to other immune cells (mean expression level) (Appendix A) and non-lymphoid tissues (mean expression level) (Appendix A). From rounds 1 and 2, we excluded 1451 and 19 genes, respectively.

All the genes selected from rounds 1 and 2 are supposed to be expressed at higher levels by tissue-resident memory T cells than by parenchymal cells. Since tissue-resident memory T cells are found at different densities in different non-lymphoid tissues, it is logical that differences in the expression of the genes in different tissues are found. However, we hypothesized that too big or too small differences between the maximum and minimum expression of a gene would indicate that the gene is constitutively expressed by parenchymal cells in a few or in many non-lymphoid tissues. Therefore, in the third round, we calculated the difference between the maximum and minimum expression of each gene in non-lymphoid tissues, and we excluded genes for which the difference was out of 2.5–8.5 log2 range (Appendix A). The range was chosen in the hypothesis that there is a difference between the highest and the lowest gene expression level due to T cell infiltration in non-lymphoid tissue more than 5.6 folds and less than 363 folds. Interestingly, the genes included in the new signature at the end of the six-step procedure were in the range 3–6 log2, corresponding to the range 8–64 folds. From round 3, we excluded two genes.

In the fourth round of selection, based on the hypothesis that all genes still present in the signature are indicative of T cell infiltration in tissues, the difference between expression in each non-lymphoid tissue and mean T cell expression (nl/Tc) was evaluated, and the mean nl/Tc (M_nl/Tc) was calculated for each tissue. If the difference between nl/Tc and M_nl/Tc ([nl/Tc]/[M_nl/Tc]) of a gene was greater than 3.32 log2 (representing a ten-fold difference), we concluded that the parenchymal cells of that tissue constitutively express the gene, and therefore, excluded it (Appendix A). In other words, the fourth round evaluated if a gene changes the level of expression in a tissue more or less than the other genes of the signature. Such behavior was considered to indicate that the gene was not expressed almost exclusively by T-cells. From round 4, we excluded four genes. Interestingly, the genes included in the new signature at the end of the six-step procedure had a mean SD [nl/Tc]/[M_nl/Tc] equal to 0.47 log2 (1.38) meaning that gene expression of these genes changes almost homogeneously across tissues, probably because they are expressed by the same cell population. On the contrary, the genes excluded from round 4 had a higher SD [nl/Tc]/[M_nl/Tc]: 1.50 log2 (2.83), 1.24 log2 (2.36), 1.21 log2 (2.31), and 1.03 log2 (2.04), respectively. The M_nl/Tc in tissue changes after the exclusion of some genes, so the same process was repeated in round 5 (Appendix A). From round 5, we excluded two genes.

Finally, in round 6, to evaluate if the selected genes were overexpressed by activated parenchymal cells (e.g., cancer cells) or activated non-T immune cells, we tested the expression of the genes in cell lines. Genes that showed less than 3.32 log2 overexpression (corresponding to ten-fold overexpression) in T cells than in cell lines from at least two different parenchymal sources and/or derived from non-T immune cell subsets were excluded (Appendix A). Through round 6, we excluded 14 genes. The six-step procedure is summarized in Figure 2A.

In summary, through this procedure we identified genes showing three main properties: (1) High expression levels by T cells and lower mean expression levels (at least 10-fold) by the other cells of the immune system purified by healthy donors, (2) high expression levels by T cells and lower mean expression levels (at least 10-fold) by non-lymphoid tissues, not showing overexpression by one/some type(s) of parenchymal cells, (3) high expression levels by T cells and lower expression levels (at least 10-fold) by tumoral and non-tumoral cell lines derived from cells other than T cells.

At the end of the six rounds of selection, the identified signature (called signature-H) included the following 15 genes: *CD2* (Gene ID: 914), *CD247* (Gene ID: 919), *CD28* (Gene ID: 940), *CD3D* (Gene ID: 915), *CD3G* (Gene ID: 917), *CD6* (Gene ID: 923), *GPR171* (Gene ID: 29909), *GZMK* (Gene ID: 3003), *ICOS* (Gene ID: 29851), *ITK* (Gene ID: 3702), *KLRB1* (Gene ID: 3820), *PYHIN1* (Gene ID: 149628), *TIGIT* (Gene ID: 201633), *TRAT1* (Gene ID: 50852)*,* and *TRBC1* (Gene ID: 28639). Most of the genes have been reported to be overexpressed by T cells by more than one paper (Table 1 and Appendix A). Despite this, signature-H is original. Indeed, no more than 47% of the genes included in the signature-H are shared with other T cell signatures, and no more than 37% of the genes included in the published signatures are shared with signature-H.

Genes belonging to signature-H show a mean 5.3 log2 (39-folds) overexpression in T cells as compared to non-immune cells (Appendix A) and a mean 5.1 log2 (34-folds) overexpression in T cells as compared to non-lymphoid tissues (Appendix A). Hierarchical clustering analysis showed that no parenchymal cells express the genes belonging to signature-H at a level similar to that of T cells (Figure 2B).

### 2.2. Comparison of Signature-H with Other T Cell Signatures for the Evaluation of Tci of Tissues from Healthy Donors

To verify whether our starting hypothesis (i.e., higher specificity correspond to better performance) was correct, we tested and compared the ability of signature-H and other known signatures to score Tci in 22 tissues from healthy donors (HT), without the use of any software to weight expression data. In particular, the mean expression of the signature genes by HT was compared with that by T cells purified from different sources, and the percentage of mRNA expression was calculated (Ts%). Therefore, Ts% of T cells is, by definition, equal to 100. Ts% provides a value allowing the comparison between T cell infiltration of two tissues and not an absolute quantization of infiltrating T cells as performed by immunohistochemical techniques. Indeed, Ts% depends on the mRNA content not considering the protein content of the genes. Moreover, mRNA levels depend on the cell activation state as well as size.

The mean Ts% in non-lymphoid tissues, as evaluated by signature-A and -D, is more than 30, which means that the mRNA signal in the human parenchyma is more than 30% of the mRNA signal of purified T cells (Figure 3A and Appendix A). Therefore, it is evident that signature-A and -D cannot be used to detect Tci without the use of dedicated software. On the contrary, the mean Ts% in non-lymphoid tissues, as evaluated by signature-B, signature-C, and signature-H, is 4.7, 5.2, and 3.0, respectively. The latter set of values seems more realistic.

Of note, the ratio of Ts% between the tissue with the highest infiltration and the tissue with the lowest infiltration was different among signature-B, signature-C, and signature-H (Appendix A). In particular, the ratio was equal to 2.2 and 3.4 with signature-B and -C, respectively, meaning that the difference of T cell infiltration among non-lymphoid organs results to be about two-fold and three-fold, respectively, when not using dedicated software. Moreover, some Ts% values resulted in being inappropriate. For example, using signature-B, Ts% of the colon (5.6) appears just higher than the Ts% of the brain (4.3) (1.3-fold difference) and breast tissue results to be less infiltrated by T cells than the brain. Similar results were observed with signature-C. On the contrary, using signature-H, the ratio of Ts% between the tissue with the highest infiltration and the tissue with the lowest infiltration was equal to 4.8 and Ts% values resulted in being more appropriate. Moreover, Ts% of the colon (4.7) is more than three folds than Ts% of the brain (1.5) and the brain and cerebellum are the tissues least infiltrated by T cells, which appears more reasonable.

The overall data analysis demonstrates that the infiltration data provided by signature-H are rather different from those provided by signature-B and signature-C.

### 2.3. Comparison of Signature-H with Other T Cell Signatures for the Evaluation of Tci of Different Types of Cancers

We compared the Tci values determined with signature-B, signature-C, and signature-H across four tumors that were considered to be infiltrated by T cells [7,10,17,19,33,34]. The mean gene expression level in each tumor sample minus the mean gene expression level in the corresponding HT, expressed as log2, was calculated for each gene, and the mean difference of all the genes included in the T cell signatures was named Ts-l2. A Ts-l2 between 0.32 and −0.32 was considered to indicate the Tci of the tumor similar to that of the corresponding HT. As shown in Figure 3B, the values derived from signature-H are indicative of a higher Tci in all four tumors as compared to the corresponding HT, while the values derived from signature-B and signature-C are indicative of a similar Tci, with the only exception of kidney clear cell cancer according to signature-C. Thus, the overall data indicate that signature-H detects high Tci in tumors known to be infiltrated by T cells, differently from signature-B and signature-C.

We used signature-H to evaluate Tci in 32 types of cancers. The Ts% for each of these cancers was calculated and compared with the mean Ts% of HT (equal to 3). If a cancer specimen showed a Ts% of 2.4−3.7, it was considered to be infiltrated at the same level as HT (low levels). Figure 3C shows that 5 and 11 cancers showed high and moderate infiltration, respectively, and that eight, seven, and one cancers showed low, very low, and extremely low infiltration, respectively. A similar analysis performed using signature-B and signature-C demonstrates that these signatures cannot detect any tumor with as high infiltration and that the number of moderately infiltrated tumors identified is lower than that identified with signature-H (Appendix A). Interestingly, five “hot” cancers (some lung cancers, stomach, and pancreas cancers) [7,35,36,37] resulted highly infiltrated with signature-H but not with signature-B and signature-C and cancers known to be infiltrated by T-cells (e.g., colon adenocarcinoma and brain neuroblastoma) resulted moderately infiltrated with signature-H but infiltrated at low levels (similar to HT) with signature-B and signature-C. In conclusion, signature-H detects several hot tumors as highly infiltrated, differently from signature-B and signature-C.

Moreover, signature-H has a higher sensitivity in the evaluation of Tci of cancer as compared the other T cell signatures. Indeed, the level of cancer Ts% determined by signature-H ranged from 1.1 (cerebellum medulloblastoma) to 8.8 (lung adenocarcinoma) (about eight-fold difference in the level of Ts%) (Figure 3C). On the contrary, the maximum difference in the Ts% of the tumors was three-fold (from 2.9 to 7.9) or about four-fold (from 2.4 to 9.2) when Ts% was calculated by signature-B or signature-C, respectively (Appendix A).

### 2.4. Comparison of Signature-H with Other T Cell Signatures for the Evaluation of Tci of Cancer Specimens

Tci evaluation of a tumor specimen may be relevant to tailoring treatment for each patient. Thus, we evaluated whether signature-H evaluates Tci of each specimen differently from signature-B and signature-C. In particular, we evaluated the percentage of cancer samples that showed a Tci absolute value (Tav, i.e., the mean expression level of the gene signature in a tissue) of the signature genes greater than the corresponding HT samples. In this case, Tav derived from the Affymetrix array platform (Tav-array) was considered. Tumor samples, for which the mean expression of the genes included in the signatures was higher than their mean +1SD expression in the corresponding HT, were considered to have a higher Tav-array than HT (T cell-infiltrated tumor samples). Figure 4A and Appendix A show that the analysis performed on four cancers using signature-H revealed a number of patients with tumor-infiltrating cells that was higher than the ones identified by signature-B and C. Notably, the spread of infiltration levels of tumor samples was wider with signature-H than other signatures, again suggesting that signature-H has a higher sensitivity as compared to signature-B and signature-C.

Next, we investigated if the patients showing the highest and the lowest Tav-array, according to signature-H, showed the highest and the lowest Tav-array, according to signature-B and signature-C too. Appendix A shows that, among the patients who were identified according to signature-H as having the highest and the lowest degree of tumor infiltration, some were not identified by signature-B, and signature-C. Figure 4B shows that, with regard to the Tav-array of kidney, clear cell adenocarcinoma, signature-B and signature-C had a concordance with signature-H of 84% (highest) and 67% (lowest) and 92% (highest) and 84% (lowest), respectively. The concordance was lower when the genes included in signature-B or signature-C and excluded by signature-H (BnoH and CnoH) were used for evaluation, while it was higher when the genes present both in signature-B and signature-H (BH) or signature-C and signature-H (CH) were used for evaluation. A similar observation was made in the case of the other cancers (not shown). Therefore, inclusion in signature-B and signature-C of genes that are poorly representative of T cells (the ones that we excluded) is the main reason why the signatures are not fully concordant with signature-H, implying that it is the specificity and not the number of genes present in the signature that is most relevant.

### 2.5. The Potential Use of Signature-H for Precision Medicine and Its Relevance for the Prognosis of Neuroblastoma Patients

We tested the usefulness of signature-H in the precision medicine perspective. Appendix A shows the levels of Ts% according to signature-H of the tumor sample of each patient in the 32 examined cancers. In several cancers, Ts% of the patient′s tumor sample ranged from a low to a very high Ts%, suggesting that signature-H may be used to evaluate Tci of patients′ cancer.

It is known that one of the independent factors for determining the prognosis of brain neuroblastoma patients is the level of Tci of the tumor, evaluated by immunohistochemistry [8]. To test the ability of signature-H to work as a prognostic tool in neuroblastoma patients, we looked for microarrays of neuroblastoma specimens linked to data concerning survival of the patients and we found a study performed on 709 specimens that used a custom oligonucleotide microarray produced by Agilent, which uses a different technology from that of Affymetrix (the one we used to set signature-H). In this array, only eight out of the 15 genes of signature-H were present (Appendix A) [38]. Since the performance in scoring Tci of signature-H containing only these eight genes (signature-H8) was similar to that of signature-H (Appendix A), we used signature-H8 to investigate the Tav-array of the signature-H8 genes of neuroblastoma specimens.

Figure 5A shows that the specimens from patients who survived less than six months had a lower Tav-array than the other specimens. The Tav-array increases with patient survival up to 4−7 years, after which it remains at the same level. Moreover, Tav-array correlates significantly with risk groups identified by the Children’s Oncology Group (COG) (Figure 5B) and tumor stage determined according to the International Neuroblastoma Staging System (INSS) (Figure 5C).

The above findings indicate that the Tci levels evaluated by signature-H8 represent an independent risk factor, as previously reported for Tci levels evaluated by histological analysis. Indeed, Tci levels evaluated by signature-H8 have prognostic value, as demonstrated by the Kaplan–Meier curves of the patients according to Tav-array (Figure 5D). Of note, patients with the lowest Tav-array (<0.125) had a death hazard ratio of 6.2 (95% CI, 2.5−9.1) as compared to patients with the highest Tav-array (>2). Moreover, children classified into the COG high-risk group can be further ranked according to the Tav-array levels, so that patients with the lowest Tav-array (<0.125) had a death hazard ratio of 2.8 (95%CI, 1.1−4.6) as compared to patients with the highest Tav-array (>2) (Figure 5E).

These results indicate that Tci determined with signature-H8 (i.e., an adapted signature-H) has prognostic value in cancers and its performance is similar to that of Tci determined by histological evaluation. Interestingly, these data suggest that the signature-H works even in evaluating Tci with platforms different from the Affymetrix platform and even when using a reduced number of genes.

### 2.6. The Potential Use of Signature-H for Predicting Response to the Anti-PD-1 Antibody Nivolumab

Predicting the response of patients to the immune checkpoint inhibitor anti-PD-1 antibody is crucial with regard to identifying patients who will not respond to the treatment. This will also save them from adverse events and reduce unnecessary expenses [18,39]. Since Tci has been correlated with the response to anti-PD-1 treatment [15,40], we investigated whether signature-H can also be used to predict the response to anti-PD-1 treatment.

We found a study on patients treated with the anti-PD-1 antibody nivolumab (Appendix A) [41]. In this study, 32 out of 38 melanoma samples were obtained from patients before starting nivolumab treatment. Twenty-six of them were analyzed by the authors through high-throughput sequencing, technology different from Affymetrix. Moreover, the expression of 14 out of the 15 genes included in signature-H was evaluated. First, we evaluated whether Tav from RNA-seq technology (Tav-RNAseq) evaluated by adapted signature-H (signature-H14) correlated with response to nivolumab (good and bad Tav-RNAseq-1 and Tav-RNAseq-2), but it did not (Appendix A).

Next, we evaluated PD-1, PD-L1, and PD-L2 expression in tumor specimens and tried to identify a combination of parameters that could predict the response to nivolumab. We found that a ligand/receptor ratio higher than 0.5 and lower than 10 was somehow predictive of the response (Appendix A). Thus, we concluded that patients with both a PD-L1/PD-1 and a PD-L2/PD-1 ratio between 0.5 and 10 belonged to the good signature-1 group (responsive to treatment) and the others belonged to the bad signature-1 group (unresponsive to treatment). The patients with good signature-1 showed a complete response (CR), partial response (PR), or progressive disease (PD) (Appendix A). The patients with bad signature-1 showed PR or PD. However, the difference between the good and bad signature-1 groups was not significant with regard to the response to the treatment and survival (Appendix A).

To improve predictability, we tested the combination of the criteria of signatures Tav-RNAseq-1 and Tav-RNAseq-2 with that of the above-described signature-1. We found that the combination of parameters PD-L1/PD-1 ratio and Tav-RNAseq-1 (signature-2) were predictive (Appendix A). To further improve the predictivity, we added a third parameter (PD-1/Tav-RNAseq ratio > 0.5). Patients who satisfied all the three parameters were included in the good signature-3 group (responsive to treatment), and the other patients were included in the bad signature-3 group (unresponsive to treatment) (Appendix A). Figure 6 shows that the overall response rate of the patients belonging to the good and bad signature-3 groups was 90% and 31%, respectively. More interestingly, the difference between the good and bad signature-3 groups was significant concerning survival (*p* = 0.0011) with a death HR equal to 7.3. Of note, if we consider only the patients who were monitored for at least 18 months, the percentage of patients who survived in the good and the bad signature-3 groups was 100% (seven out of seven) and 15% (two out of 13), respectively. The three criteria chosen on a mathematical basis may also have a biological meaning, as discussed in the Discussion Section.

The results give the proof of principle that signature-H can be used to improve the prediction of response to checkpoint inhibitors.

## 3. Discussion

In the present study, we identified a new T cell signature (signature-H). Most of the 1507 genes analyzed in this study were expressed by T cells, and several were more expressed in T cells than other types of cells. However, in order to create signature-H, we selected genes through multiple rounds of analysis applying very stringent criteria (Figure 2A). These steps left us with 15 genes that were truly representative of T cells in tissues from both healthy and diseased subjects, as shown in Figure 2B. Signature-H is original and, as expected, evaluates T cell infiltration differently from the other T cell signatures, at least when not using dedicated software. This is demonstrated by the different T cell infiltration levels detected by signature-H and the other signatures in HT (Figure 3A), in cancer types (Figure 3B,C and Appendix A) and in cancer specimens (Figure 4, Appendix A).

Some reasoning may suggest that signature-H is more sensitive and predictive than signature-B and signature-C in the evaluation of T cell infiltration, at least when a dedicated software is not used. (1) signature-H describes brain and cerebellum as the least infiltrated tissues by T cells among the non-lymphoid HT whereas signature-B and signature-C do not, (2) the difference between T cell infiltration levels in the most and the least infiltrated non-lymphoid tissues results to be higher with signature-H than signature-B and signature-C. In our opinion, an about five-fold difference (signature-H) is more reasonable than about a two- to three-fold difference (signature-B), (3) four tumors considered to be infiltrated by T cells [7,10,17,19,33,34] resulted to be more infiltrated by T-cells as compared to the corresponding HT using signature-H and resulted to be infiltrated at the same HT level using signature-B and signature-C, with the only exception of kidney clear cell cancer according to signature-C (Figure 3B), (4) cancers known to be highly infiltrated (“hot” cancers) [7,35,36,37] are described “hot” by signature-H and not by signature-B and signature-C (Appendix A), (5) some cancers (e.g., brain neuroblastoma and colon adenocarcinoma) considered to be infiltrated by T cells [7,8,14] showed moderate Tci according to signature-H and had low Tci according to signature-B and signature-C.

The higher sensitivity of signature-H was also useful for ranking patients with the same tumors according to Ts% (Figure 4, Appendix A). Interestingly, the percentage of melanomas previously described to be infiltrated by T cells is about 60% [15]. This is coherent with our results (Appendix A), demonstrating that the same percentage of melanomas has a Ts% that is higher than the mean Ts% of HT.

In our opinion, the possible higher sensitivity of signature-H as compared to the other signatures is determined by the way we selected the genes of the signature. In fact, the expression levels of each gene included in signature-H are higher in T cells as compared to non-lymphoid tissues (Figure 2B), meanwhile some genes included in signature-B and signature-C are expressed at similar levels in T cells and some non-lymphoid tissues (Appendix A). Interestingly, the absence of some of the genes included in signature-H in custom arrays does not preclude the use of a modified signature-H (Appendix A and not shown). On the contrary, in signature-B and signature-C, some genes work better than others (Figure 4B).

The induction of patients’ immune response against neuroblastoma is one of the mechanisms that contribute to the frequently observed spontaneous regression of this cancer [42]. Signature-H, contrary to signature-B and signature-C, demonstrates that neuroblastomas have a higher level of infiltration than the brain from healthy donors (Figure 3B) and more than the mean infiltration of HT (Figure 3C). Infiltration of T cells, as evaluated by immunohistochemistry, has been associated with favorable clinical outcomes in recent studies [8,43]. In particular, Mina et al. demonstrated that 50% of the patients with the highest T cell density had a significantly higher survival rate than the patients with the lowest T cell density. When signature-H was used to evaluate the Tci of 709 neuroblastoma specimens, the results were comparable with those obtained by Mina et al. (Figure 5). This finding confirms that signature-H is a valid tool for investigating Tci in cancers.

Immune checkpoint inhibitors have been reported to have impressive outcomes, but a few patients respond to this treatment [19,20,44]. There is, therefore, a need to identify markers that can predict the response of patients to treatment with immune checkpoint inhibitors. Several predictive markers have been suggested, but their predictive power is low [18,39]. Therefore, we tried to predict patients′ response to anti-PD-1 treatment based on three parameters having a biological plausibility: (1) PD-L1/PD-1 ratio between 0.5 and 10 means that a patient can respond to nivolumab if the tumor expresses a sufficient amount of PD-1 ligand (but not too much) able to trigger PD-1-signaling in T cells, (2) a Tav-RNA-seq value greater than 1 (representing Tci) means that a patient can respond to nivolumab if the tumor is infiltrated by T cells, and (3) a PD-1/Tav-RNA-seq value ratio greater than 0.5 means that a patient can respond to nivolumab if T cells are exhausted (thus expressing a sufficient amount of PD-1) (Figure 6). Of note, the findings showed that adding the Tav-RNA-seq and PD-1/Tav-RNA-seq values was crucial for the predictability of treatment response. The findings confirm that Tci and the density of PD-L1 on tumor are crucial for the response to mAb interfering with PD-1 pathway [15] and suggest that Tci evaluated by signature-H may be used to predict response to the treatment.

The signature approach is burdened by considerable limits. The main limitation is the signatures give information on genes which are mainly, but not exclusively, expressed by a cell subset (e.g., T cells). In the case of T cell signatures, T cells are considered having a homogeneous phenotype, but the truth is they are formed by diver subsets that may be either activated or not, and that can be present at different percentages in different tissues. Our signature-H considers T cell activation by including some genes that are overexpressed by activated T cells. However, the abundance of a gene does not provide percentages of T lymphocytes present in a specific tissue. For this reason, all T cell signatures, do not reflect the real situation in vivo, but they aim at evaluating the relative Tci levels in comparison to other Tci levels derived from healthy tissues and/or tumors. From this aspect, the accuracy of the transcriptional analysis is lower, at least in theory (i.e., not considering technical and statistical bias), than the accuracy provided by immunohistochemistry analysis. This is also due to the fact that gene signatures evaluate mRNA expression levels which not always are good indicators for protein expression. Moreover, it is well known that immunohistochemistry is so far the only technique providing quantitative information on the distribution and composition of the cell infiltrate.

The limitations of signature-H are three: (1) The six-step procedure eliminated from the signatures most of the genes expressed at high levels by non-T cell subsets, including non-T cell activated subsets (genes expressed at high levels in non-T cell activated subsets are excluded by step 6. However, genes expressed by signature-H are expressed by cells different from T cells and it may happen that one of the genes is overexpressed by a small non-T cell subset. For example, some of the genes included in the signature (e.g., *GZMK* and *KLRB1*) are expressed by natural killer (NK) cells. Therefore, in a tissue massively infiltrated by NK cells, Tci value from signature-H may be slightly higher than expected. (2) In order to exclude genes expressed by activated non-T cells and tumor cells, we considered the level of expression of the genes by non-T cell lines (both tumor-derived and immortalized). Although cell lines may be considered activated T cells, their phenotype does not overlap completely either with the phenotype of activated cells, or with the phenotype of tumor cells. Therefore, some genes expressed by activated non-T cells or tumors derived by non-T cells may not be excluded by the signature. (3) We did not test signature-H in an inflammatory context. There is no reason why it would not work, but this is a point we want to explore in future studies. However, despite the limitations encountered using signature-H, we demonstrated that signature-H works quite well. Indeed, it describes as infiltrated by T cells cancers known to be infiltrated by T cells, and Tci derived by signature-H correlates with the survival rate for neuroblastoma determined by immunohistochemical analysis.

The six-step selection method that we used could be applied to set up signatures of subsets of immune cells (such as, for example, CD8^+^ and CD4^+^ T cells, and B cells) that do not share almost all genes with other cell subsets. On the contrary, this method cannot be applied to set up a signature of cell subsets sharing almost all genes with other subsets even if at slightly different levels of expression. In our opinion, this is the main limitation of cell signatures as compared to flow cytometry and immunohistochemistry.

In conclusion, our study presents a new T cell signature that excludes from earlier signatures genes that are not overexpressed exclusively by T cells as compared to non-T cells, parenchymal cells included. Our results suggest that evaluation of Tci by signature-H using microarrays (as well as high-throughput sequencing) may help characterize tumor biology, provide prognostic information, and estimate how likely a tumor is to respond to treatments. This leads us to hypothesize that, for certain tumors, signature-H may represent an alternative approach to immunohistochemistry or may complement immunohistochemistry providing additional information. Evaluation of mRNA expression would be easier to standardize and less time-consuming than immunohistochemistry, even if a standardization for each platform might be necessary. However, large-scale studies are necessary to evaluate whether the proposed method can be applied in the clinical context in alternative or support of immunohistochemistry.

## 4. Materials and Methods

### 4.1. Data Source, Tumor Types, and Control Samples

We used the Genevestigator V3 suite (NEBION AG, Zurich, Switzerland) [30], through which normalized absolute values of gene expression generated using the Affymetrix Human Genome U133 Plus 2.0 platform were downloaded. The microarray data in Genevestigator are normalized at two levels: Robust Multi-array Average (RMA) within experiments (through the Bioconductor package “affy” and a customized version of the package “affyExtensions”) and trimmed mean adjustment to a target for normalization between datasets. With regard to the latter, the trimmed mean is determined by calculating the mean of all the expression values in an experiment (across all samples) after excluding the top 5% and the bottom 5%. The combination of the two levels makes data highly comparable across different experiments, thus making it possible to pool data without further normalization. A few genes (e.g., *TRA*, *MGC40069*, and *AV8S2*) were not available in the used platform, therefore, we could not evaluate them.

Gene expression data were obtained from datasets that are publicly available from Gene Expression Omnibus [31] and the European Bioinformatics Institute [32].

The Genevestigator database was queried in March 2018. We extracted and considered data from 16602 arrays from healthy and diseased tissue and cell lines (specifically, 462 arrays from purified human T cells and T cell subsets, 594 arrays from four purified human cell subsets of non-activated (main progenitors and mature cells) and activated immune non-T cells, specifically B cells, dendritic cells, granulocytes, mono/macrophages, 5990 arrays from 152 types of tissues from healthy donors, 2549 arrays from 474 non-T cell lines, 7007 arrays from 32 types of cancers of patients). In subsequent analyses, the tissues from healthy donors were merged into 22 categories. The arrays from each type of tumor-derived from different datasets were normalized with the Genevestigator software.

We included in the analysis the arrays on mRNA samples extracted from purified T cells, purified non-T immune cells, and healthy tissues that: (1) Were taken from healthy donors, and (2) were not subjected to in vitro experimental treatments. We also included in the analysis the arrays on mRNA samples extracted from non-T cell lines that were not subjected to experimental treatments. We further included in the analysis the arrays on mRNA samples extracted from cancer specimens that: (1) were primary tumors, (2) were not obtained by laser capture microdissection, and (3) were not subjected to experimental treatments.

For the study of neuroblastomas, data from the ArrayExpress database of the European Bioinformatics Institute (dataset E-MTAB-1781) were downloaded. The dataset included data from 709 neuroblastoma specimens (detailed information is provided in Appendix A).

For evaluating the response to nivolumab treatment, data from Gene Expression Omnibus (dataset GSE78220) were downloaded. The dataset included data from 26 melanoma specimens (detailed information is provided in Appendix A).

### 4.2. T Cell Infiltration Scores Definition

To estimate T cell infiltration we considered three different scores. The purpose of each score is to calculate Tci of a tissue in relation to the Tci of another/other tissue(s). For clarity, the different scores are defined below:

Tci score percentage (Ts%) estimates the percentage of T cells infiltration in healthy or cancerous tissues. Ts% provides a value allowing a relative quantization of infiltrating T cells. For example, if Ts% of tissue X = 2 and Ts% of tissue Y = 1, it means that tissue X has two times more infiltrating T cells than tissue Y. Ts% was calculated under the assumption that purified T cells have a Ts% = 100. Consequently, Ts% can be calculated only if data on purified T cells are available in a platform. Interestingly, following Ts% calculation, Ts% value allows comparison across different platforms.

Tci absolute value (Tav) is the mean expression level of the gene signature (log2) in a tissue (healthy or cancerous) and estimates the density of T cells within a tissue. Analogous to Ts%, Tav provides a value allowing a relative quantization of infiltrating T cells, and can be calculated even if data on purified T cells are not available in a platform. However, Tav is platform-specific, consequently, the Tav of a tissue cannot be compared with other Tavs derived from different platforms. For this reason, Tav is followed by the platform name that has been used (such as Affymetrix array, and RNAseq).

Tci score log2 (Ts-l2) is the difference between the mean expression levels of the gene signature (log2) in a cancerous tissue and the mean expression levels of the gene signature (log2) in the corresponding tissue from healthy donors (HT). Consequently, Ts-l2 can be calculated only if data on both cancerous and the corresponding HT are available. Interestingly, following Ts-l2 calculation, Ts-l2 value allows comparison across different platforms.

### 4.3. Quantification and Statistical Analysis

Hierarchical clustering was performed using the hierarchical clustering tool of the Genevestigator suite under the conditions of Euclidean distance and optimal leaf-ordering.

The Student *t*-test (unpaired *t*-test with Welch′s correction) was used for analyzing differences between the two groups.

In the analysis of neuroblastoma specimens, the correlation of Tav from Affymetrix array (Tav-array), with survival time, risk according to the Children′s Oncology Group (COG) system, and disease stage according to the International Neuroblastoma Staging System (INSS) was compared using one-way ANOVA. For the comparison of Kaplan–Meier curves, the log-rank Mantel–Cox test was used. The hazard ratio (95% CI) was calculated using the log-rank method. For comparison between good and bad signatures, the contingency chi-square test was used. Pearson correlation analysis was used in the calculation of correlation coefficients and their significance.

All statistical analyses were conducted using the Prism software (GraphPad Software, La Jolla, CA, USA).

## Figures and Tables

**Figure 1 ijms-20-05242-f001:**
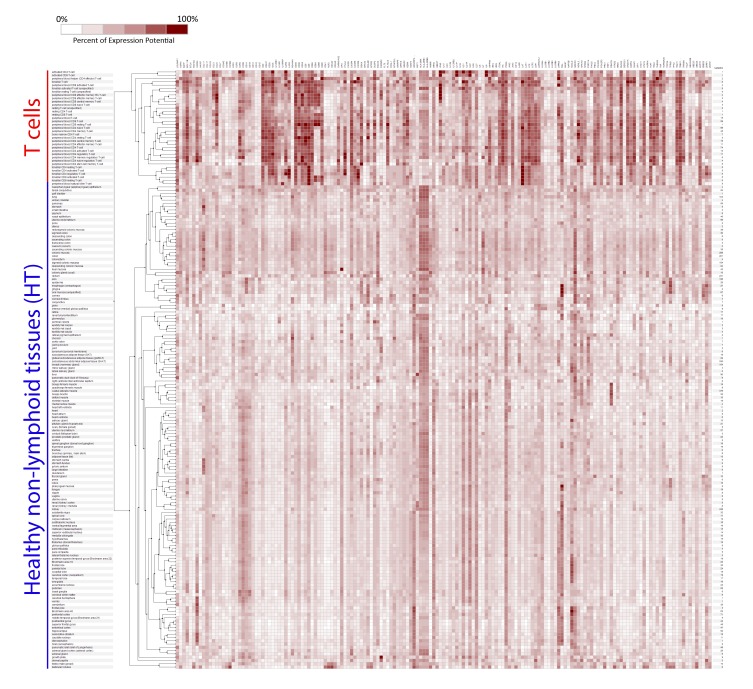
Expression of the genes included in published T cell signatures. Expression of the genes included in the T cell signatures reported by other studies [22,23,24,25] was evaluated in T cells and in non-lymphoid tissues from healthy human donors (blue) by using the hierarchical clustering tool.

**Figure 2 ijms-20-05242-f002:**
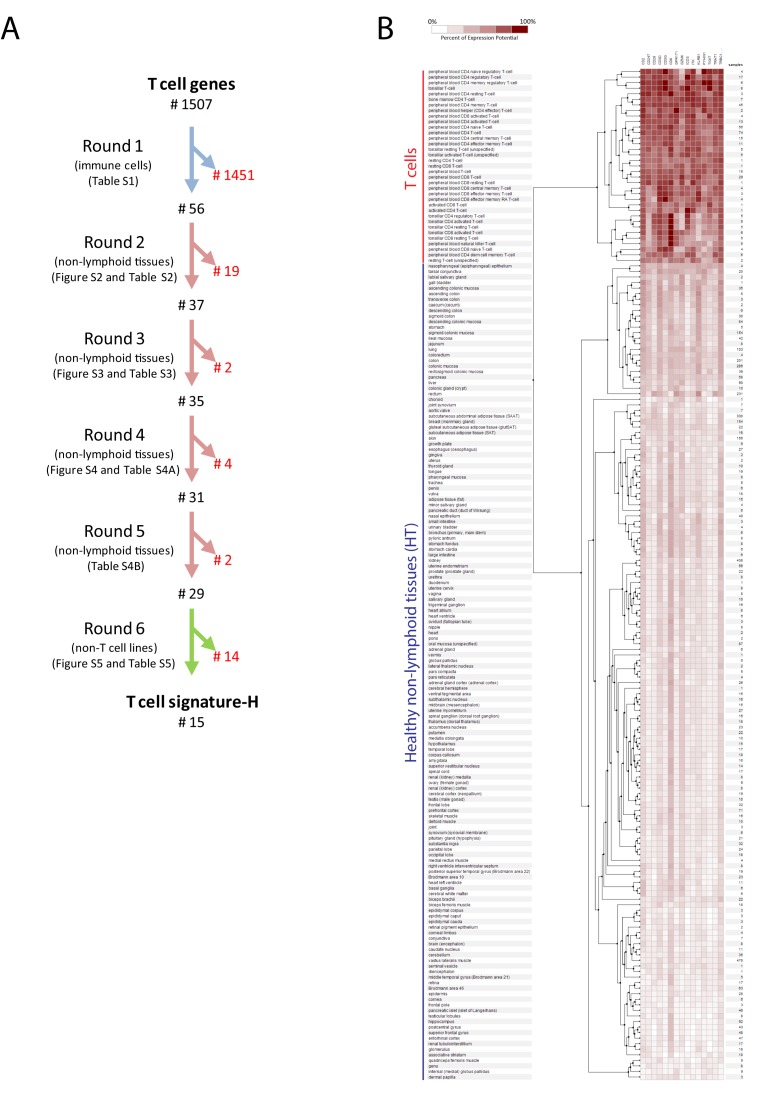
Selection of genes for the new T cell signature (signature-H) and first validation test. (**A**) Summary of the six-step selection method used to develop signature-H. We evaluated 1507 genes reported to be overexpressed by T cells or by T cell subsets [22,23,24,25,27,28,29] and excluded the genes that did not meet the set criteria. The number of genes excluded (red characters) and saved (black characters) after each round, the cells/tissues used for the selection (blue arrows, immune cells other than T cells; pink arrows, HT; green arrow, non-T cell lines) and the tables/figures showing the criteria used in each round are specified. (**B**) Expression of the genes included in signature-H in T cells and in non-lymphoid tissues from healthy human donors (HT) by using the hierarchical clustering tool.

**Figure 3 ijms-20-05242-f003:**
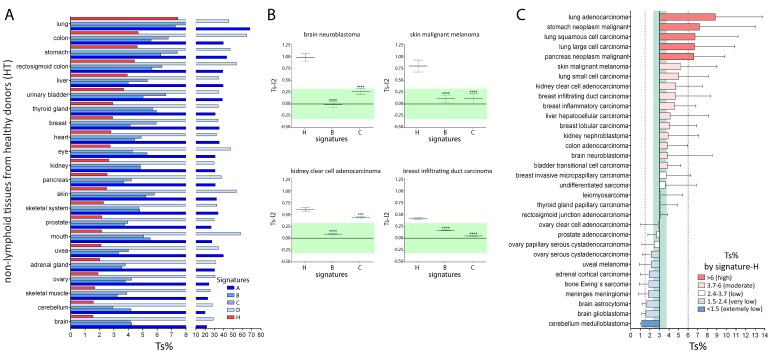
Tci score% (Ts%) of HT and cancers determined with signature-H and previously published T cell signatures. (**A**) The expression of genes of each signature in 22 HT types was divided by the mean expression of the genes in T cells, and value was expressed as a percentage (Tci score%–Ts%). (**B**) The Tci score log2 (Ts-l2) of each cancer was calculated as the mean expression of the signature genes in each cancer specimen minus the mean expression of the same genes in the corresponding HT. The mean ± 1SE Ts-l2 is reported. Ts-l2 between 0.32 and −0.32 (green area corresponding to 80–125% value of HT) indicates that the Ts-l2 of the tumor is similar to that of the corresponding HT. Student *t*-test: *** *p* < 0.001, **** *p* < 0.0001, signature-B and signature-C vs. signature-H. (**C**) The mean ± 1SD expression of genes of signature-H in 32 types of cancers was divided by the mean expression of the genes in T cells, and Ts% was plotted in comparison to the mean Ts% of HT, which was equal to 3.0. The green area corresponds to 80−125% of the Ts% of HT.

**Figure 4 ijms-20-05242-f004:**
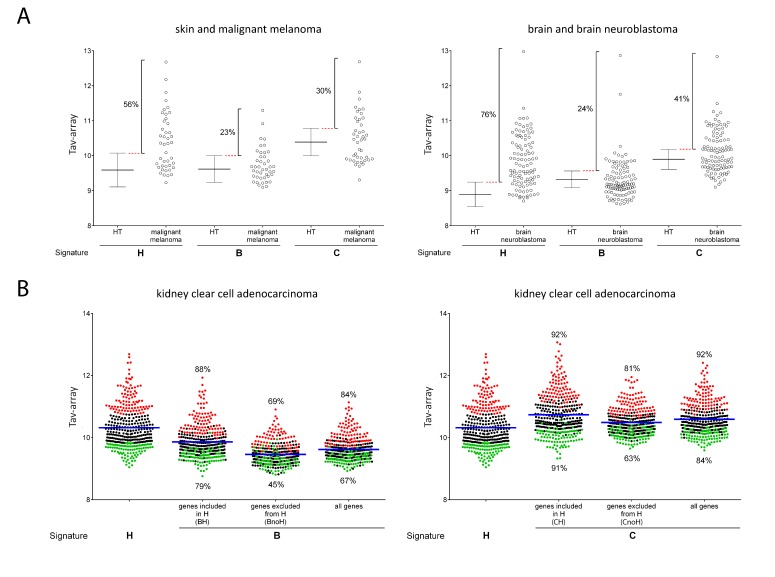
Tci absolute array value (Tav-array) of cancer samples determined with signature-H and previously published T cell signatures (**A**) Tav-array according signature-H, signature-B, and signature-C in cancer specimens and HT specimens from the corresponding tissue is shown. Cancer specimens for which the Tav-array is higher than the mean + 1SD of the Tav-array in HT were considered to be infiltrated by T cells and their percentage is reported. (**B**) The Tav-array in kidney clear cell adenocarcinoma according to signature-H and signature-B (left panel) and signature-H and signature-C (right panel) is shown. Specimens were ranked according to the Tav-array belonging to signature-H. The specimens belonging to the top (Q1) and bottom (Q4) quartile are shown in red and green, respectively. In each panel, the Tav-array from signature-B or signature-C (all genes), the Tav-array derived from genes belonging to both signature-B (or signature-C) and signature-H (BH and CH, respectively), and the Tav-array derived from genes belonging to signature-B (or signature-C) and absent in signature-H (BnoH and CnoH, respectively) were calculated for each specimen. The specimens belonging to Q1 and Q4 according to signature-H are in red and green, respectively. The percentage of specimens in Q1 and Q4 according to signature-H that were still in Q1 and Q4 according to signature-B (or signature-C), BH (or CH) and BnoH (or CnoH) is reported. The mean Tav-array according to the signatures is also reported (blue line).

**Figure 5 ijms-20-05242-f005:**
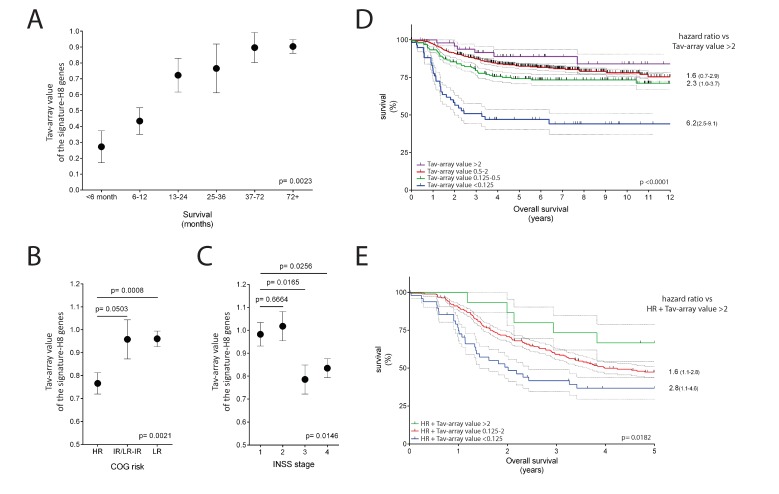
Prognostic value of signature-H in neuroblastoma. (**A**) Mean ± 1SE expression level of the signature-H8 genes in neuroblastoma specimens (Tci absolute array value–Tav-array) from patients ranked by survival. (**B**,**C**) Tav-array (mean ± 1SE) in neuroblastoma specimens from patients grouped according to risk groups identified by the Children′s Oncology Group (COG) (**B**) and stages according to the International Neuroblastoma Staging System (INSS) (**C**). In panels (**A**–**C**), one-way ANOVA was used for comparing the Tci absolute array values across the groups and survival values (right). Student′s *t*-test was used for comparing the Tav-array across groups. (**D**) Kaplan–Meier curve of patients ranked into four groups according to the Tav-array of the corresponding tumor specimen. (**E**) Kaplan–Meier curve of the COG HR patients ranked into three groups according to the Tav-array of the corresponding tumor specimen. In panels (**D**,**E**), the log-rank Mantel–Cox test was used for comparing survival curves. The hazard ratio (95%CI) was calculated with the log-rank method and is shown on the right.

**Figure 6 ijms-20-05242-f006:**
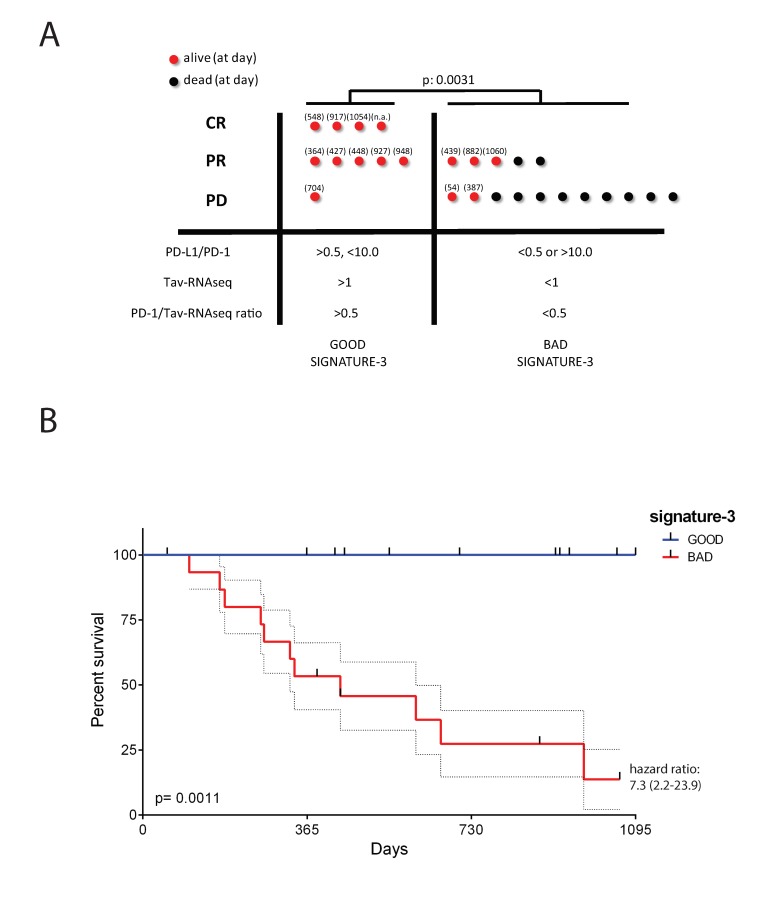
Prediction of response to nivolumab via a multiparametric tool that includes values from signature-H. (**A**) The mean expression level of the genes belonging to signature-H14 (Tci absolute RNAseq value - Tav-RNAseq) in melanoma specimens from the study by Hugo et al. (Appendix A) was evaluated, and the PD-1/Tav-RNAseq ratio was calculated. The parameter PD-1/Tav-RNAseq ratio >0.5 was combined with those described in Appendix A and used to divide patients into two groups: Good signature-3 (specimens with all parameters satisfied) and bad signature-3 (specimens with one or more parameters unsatisfied). The dots represent single patients (the red dots indicate living patients and the black dots indicate dead patients) classified according to response to treatment (complete response, CR; partial response, PR; progressive disease, PD). The numbers in parentheses refer to the number of days between diagnosis and the last follow-up. The difference between the groups was evaluated by the contingency chi-square test. (**B**) The Kaplan–Meier curves for the good and bad signature-3 groups are shown. The response of the patients was evaluated by the log-rank Mantel–Cox test. The hazard ratios (95%CI) calculated with the log-rank method are shown.

**Table 1 ijms-20-05242-t001:** Genes included in the newly set T cell signature (signature-H) and their inclusion in the published signatures of T cells and T cell subsets. X: presence.

**Name of the Signature in the Manuscript**		**Signature-A**	**Signature-B**	**Signature-C**	**Signature-D**	**/**	**/**	**/**	**Signature-H**	**Number of Signatures in Which the Gene is Overexpressed (Signature-H Excluded)**
**References**		[22]	[23]	[24]	[25]	[27]	[28]	[29]	the present study	
**Populations Investigated by the Study**		T cells	T cells	T cells	T cells	T cells subpopulations	T cells subpopulations	T cells subpopulations	T cells	
**Number of the Genes Included in the Signature**		86	17	19	76	341	105	1002	15	
**Gene**	***CD2***			X				X	X	**2**
	***CD247***	X						X	X	**2**
	***CD28***	X	X	X			X	X	X	**5**
	***CD3D***	X	X	X	X	X	X	X	X	**7**
	***CD3G***	X	X	X	X	X	X	X	X	**7**
	***CD6***		X	X	X	X	X	X	X	**6**
	***GPR171***							X	X	**1**
	***GZMK***				X	X		X	X	**3**
	***ICOS***		X		X		X	X	X	**4**
	***ITK***	X				X	X	X	X	**4**
	***KLRB1***							X	X	**1**
	***PYHIN1***							X	X	**1**
	***TIGIT***					X			X	**1**
	***TRAT1***		X	X		X	X	X	X	**5**
	***TRBC1***			X					X	**1**

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
