# Peer review of "Identification of 15 T Cell Restricted Genes Evaluates T Cell Infiltration of Human Healthy Tissues and Cancers and Shows Prognostic and Predictive Potential"

_ijms, 2019, doi:10.3390/ijms20205242_

Round 1

Reviewer 1 Report

A manuscript “Identification of 15 T-cell restricted genes evaluates T-cell infiltration of human healthy tissues and cancers and shows prognostic and predictive potential” by Cari L, et al, is devoted to a study in silico on T-cells, infiltrating non-lymphoid human normal and cancer tissues. The authors propose the new panel to determine the T-cell “signature”, performing 6 rounds of enrichment and exclusion of the candidate genes. In result, the authors propose the signature “H” for the tissue infiltrating T-cells, consisting of 15 genes.

The authors conclude that, using microarrays and/or high throughput sequencing and evaluation of the expression levels of genes of the signature “H”, it is possible to predict the course of disease and response to treatment. The authors propose substitute an immunohistochemistry analysis by microarray or sequencing technique.

Also, authors thin that it is easier to standardize the data on mRNA expression, obtained by abovementioned methods, than by routine immunohistochemistry.

This paper is interesting, but conclusions seem to be not well argumented. I have the following questions and comments to the authors:

The authors analyzed the relative expression of the genes. It was not the absolute expression, and the authors worked with the data on mRNAseq and also obtained by hybridization of microarrays. Of course, these data are quite heterogeneous, because the calculations are greatly dependent on the method used and qualification of the bioinformatician. You have to keep in mind, that data represent a crude expression of all cells, and not only infiltrating T-cells. In the tissue, especially in tumor cells, you can find many macrophages and other hematopoietic cells as well.

Even if you have as a control isolated from peripheral blood B- and T-cells, you can’t relate the strength of expression signal, because you have no clue how many cells infiltrated tissue.

Also, T-cells in the tumor tissue are activated very often, and they would have different expression pattern from the normal T-cells from the peripheral blood. That’s why I do not think that the obtained data in silico reflect the real situation in vivo.

In the light of discussed above, three main properties of the selected genes, according to the authors, namely, “1) high expression levels by T cells and lower mean expression levels (at least 10-fold) by the other cells of the immune system purified by healthy donors; 2) high expression levels by T cells and lower mean expression levels (at least 10-fold) by non-lymphoid tissues, not showing overexpression by one/some type(s) of parenchymal cells; 3) high expression levels by T cells and lower expression levels (at least 10-fold) by tumoral and non-tumoral cell lines derived from cells other than T cells” can not be considered as such. Moreover, tumor cell lines differ essentially from the tumors, due to in vitro selection and growth conditions; there is no influence of microenvironment either.

The fast analysis of few out of 15 selected genes (CD2, CD247, CD28, CD3D, CD3G, CD6, GPR171, GZMK, ICOS, ITK, KLRB1, PYHIN1, TIGIT, TRAT1, and TRBC1) allow us to conclude on an important role of inflammation, what was not considered by the authors in the present work. For example, GZMK, granzyme K, is expressed mainly by natural killer (NK) and cytotoxic T-cells (CTLs). KLRB1 is also expressed by NK cells. ICOS is upregulated on activated T-cells and induces IL2 production. PYHIN1 is interferon-inducible protein, and was found upon stimulation of B-cells by INF. Moreover, PYHIN1 is a subunit of inflammasome.

I mean, that the authors got the signature of the immune response rather that simply of tissue infiltrating T-cells. I am sure that in a Discussion part these thoughts should be added.

Another small remark – the genes should be called, according to international nomenclature, and do not have the trivial names. Alternatively, the accession number could be added to avoid confusion.

Concerning the author statement that it is easier to standardize the data on mRNA expression, than by routine immunohistochemistry, I totally disagree, because by use of the same antibody the different labs can produce similar results. And, importantly, you can see different T-cell subsets, what you can not do by crude sequencing.

Concerning suggestion of using the “H” signature in clinical settings, it does not seem feasible for the nowadays. The cost of antibodies, automated machine for staining and automated signal reader is lower, compared to RNAseq procedures and analyzing of the data.

Overall, I would suggest the authors to carefully revise their paper.

Author Response

Authors:

We were pleased to have an opportunity to revise our manuscript entitled “Identification of 15 T-cell restricted genes evaluates T-cell infiltration of human healthy tissues and cancers and shows prognostic and predictive potential” and we thank the Reviewer for the suggestions that improved our manuscript.We hope that the revised manuscript will meet the expectations of the Reviewer.

Reviewer:

A manuscript “Identification of 15 T-cell restricted genes evaluates T-cell infiltration of human healthy tissues and cancers and shows prognostic and predictive potential” by Cari L, et al, is devoted to a study in silico on T-cells, infiltrating non-lymphoid human normal and cancer tissues. The authors propose the new panel to determine the T-cell “signature”, performing 6 rounds of enrichment and exclusion of the candidate genes. In result, the authors propose the signature “H” for the tissue infiltrating T-cells, consisting of 15 genes.

The authors conclude that, using microarrays and/or high throughput sequencing and evaluation of the expression levels of genes of the signature “H”, it is possible to predict the course of disease and response to treatment. The authors propose substitute an immunohistochemistry analysis by microarray or sequencing technique.

Also, authors think that it is easier to standardize the data on mRNA expression, obtained by above mentioned methods, than by routine immunohistochemistry.

This paper is interesting, but conclusions seem to be not well argumented.

Authors:

We thank Reviewer for finding our paper interesting and appreciate the efforts made to improve the manuscript, in particularly, the Discussion Section. We have responded in a point-by point fashion. Line numbers are referred to the revised version of the manuscript.

Reviewer:

I have the following questions and comments to the authors:

The authors analyzed the relative expression of the genes. It was not the absolute expression, and the authors worked with the data on mRNAseq and also obtained by hybridization of microarrays. Of course, these data are quite heterogeneous, because the calculations are greatly dependent on the method used and qualification of the bioinformatician. You have to keep in mind, that data represent a crude expression of all cells, and not only infiltrating T-cells. In the tissue, especially in tumor cells, you can find many macrophages and other hematopoietic cells as well.

 Authors:

Regarding the heterogeneity of data, we can’t agree with Reviewer because they do not depend on bioinformaticians but on the software (a largely used and validated software platform- please see ref 30 of the revised manuscript - that provides researches with the opportunity to use and consult datasets obtained from several laboratories) and the method (in this case the use of 15 genes included in the signature-H). However, we agree with Reviewer that “data represent a crude expression of all cells, and not only infiltrating T-cells. In the tissue, especially in tumor cells, you can find many macrophages and other hematopoietic cells as well.” This is a clear intrinsic limitation of our study, a limitation shared with all the published signatures identified by using bioinformatic tools. Therefore, we have discussed the limitations of gene signature selection in the Discussion Section (lines 434-66 of the revised manuscript).

Reviewer:

Even if you have as a control isolated from peripheral blood B- and T-cells, you can’t relate the strength of expression signal, because you have no clue how many cells infiltrated tissue.

Authors:

We agree with the Reviewer about the issue of the strength of expression signal is another limitation that needs to be considered during the gene signature selection process. We have discussed this point in the Discussion Section (lines 445-48 of the revised manuscript) focusing on the difference between mRNA and protein expression.

Reviewer:

Also, T-cells in the tumor tissue are activated very often, and they would have different expression pattern from the normal T-cells from the peripheral blood. That’s why I do not think that the obtained data in silico reflect the real situation in vivo.

Authors:

We agree entirely that “T-cells in the tumor tissue are activated very often”. For this reason, genes expressed at higher levels by activated T cells (e.g. GPR171, ICOS, and TIGIT), are included in signature-H (lines 438-39 of the revised manuscript). Moreover, we believe that the in silico data cannot “reflect the real situation in vivo” as clearly stated in the revised version of the manuscript (lines 440-42 of the revised manuscript). However, although signature-H does not “reflect the real situation in vivo”, the results we have obtained using signature-H are comparable to those described by the literature obtained with non-bioinformatic techniques (lines 463-66 of the revised manuscript).

Reviewer:

In the light of discussed above, three main properties of the selected genes, according to the authors, namely, “1) high expression levels by T cells and lower mean expression levels (at least 10-fold) by the other cells of the immune system purified by healthy donors; 2) high expression levels by T cells and lower mean expression levels (at least 10-fold) by non-lymphoid tissues, not showing overexpression by one/some type(s) of parenchymal cells; 3) high expression levels by T cells and lower expression levels (at least 10-fold) by tumoral and non-tumoral cell lines derived from cells other than T cells” can not be considered as such. Moreover, tumor cell lines differ essentially from the tumors, due to in vitro selection and growth conditions; there is no influence of microenvironment either.

Authors:

We believe we demonstrate signature-H shows all the 3 properties mentioned above. Our statement is not based on physiopathological considerations but on mathematical unbiased evidence. The mathematical evidence derives from the use of a six-step procedure using 3.32 log2 as a cut-off value, representing a 10-fold overexpression.

Despite this point, we agree with the Reviewer concerning the differences across cell lines, tumors, and activated cells. This is another limitation of the study that has been reported in the Discussion Section (lines 456-61 of the revised manuscript).

Reviewer:

The fast analysis of few out of 15 selected genes (CD2, CD247, CD28, CD3D, CD3G, CD6, GPR171, GZMK, ICOS, ITK, KLRB1, PYHIN1, TIGIT, TRAT1, and TRBC1) allow us to conclude on an important role of inflammation, what was not considered by the authors in the present work. For example, GZMK, granzyme K, is expressed mainly by natural killer (NK) and cytotoxic T-cells (CTLs). KLRB1 is also expressed by NK cells. ICOS is upregulated on activated T-cells and induces IL2 production. PYHIN1 is interferon-inducible protein, and was found upon stimulation of B-cells by INF. Moreover, PYHIN1 is a subunit of inflammasome.

I mean, that the authors got the signature of the immune response rather that simply of tissue infiltrating T-cells. I am sure that in a Discussion part these thoughts should be added.

Authors:

We agree with the Reviewer about the inclusion in the signature of genes expressed by NK cells and activated T cells. This is another limitation of the study that has been discussed in the Discussion Section (lines 449-56 of the revised manuscript).

Reviewer:

Another small remark – the genes should be called, according to international nomenclature, and do not have the trivial names. Alternatively, the accession number could be added to avoid confusion.

Authors:

We have inserted Gene IDs of the genes belonging to signature-H in the Results Section, paragraph 2.1 (lines 134-37 of the revised manuscript).

Reviewer:

Concerning the author statement that it is easier to standardize the data on mRNA expression, than by routine immunohistochemistry, I totally disagree, because by use of the same antibody the different labs can produce similar results. And, importantly, you can see different T-cell subsets, what you can not do by crude sequencing.

Concerning suggestion of using the “H” signature in clinical settings, it does not seem feasible for the nowadays. The cost of antibodies, automated machine for staining and automated signal reader is lower, compared to RNAseq procedures and analyzing of the data.

Authors:

We have changed the statements regarding  the use of immunohistochemistry as a prognostic tool and the application of signature-H in clinical settings in the revised version of the manuscript (lines 443-48 and 471-82 of the revised manuscript).

Reviewer:

Overall, I would suggest the authors to carefully revise their paper.

Authors:

We hope that the revised manuscript will meet the expectations of the Reviewer.

Reviewer 2 Report

In this work, the authors used a “selective subtraction” strategy to select 15 T-cell –specific genes as a new molecular signature (signature-H) and found this signature has a higher sensitivity a better predictivity in evaluation of T-cell infiltration in healthy tissues as well as some cases of cancers. Based on their analysis, the authors suggested that signature-H evaluates T-cell infiltration levels of tissues may be used as a prognostic tool in the precision medicine perspective after an appropriate clinical validation. The experiments were well- and logically-designed and the data assessments seemed to be appropriate. The manuscript is well written and is ready for publication. Reading the manuscript, only one small grammatical error is sported out (the first sentence of the abstract).

Author Response

Authors:

We were pleased to have an opportunity to revise our manuscript entitled “Identification of 15 T-cell restricted genes evaluates T-cell infiltration of human healthy tissues and cancers and shows prognostic and predictive potential”. We thank the Reviewer for the positive evaluation of our work.

Reviewer:

In this work, the authors used a “selective subtraction” strategy to select 15 T-cell –specific genes as a new molecular signature (signature-H) and found this signature has a higher sensitivity a better predictivity in evaluation of T-cell infiltration in healthy tissues as well as some cases of cancers. Based on their analysis, the authors suggested that signature-H evaluates T-cell infiltration levels of tissues may be used as a prognostic tool in the precision medicine perspective after an appropriate clinical validation. The experiments were well- and logically-designed and the data assessments seemed to be appropriate. The manuscript is well written and is ready for publication. Reading the manuscript, only one small grammatical error is sported out (the first sentence of the abstract).

Authors:

The grammatical error has been amended.

Round 2

Reviewer 1 Report

A manuscript became better, and more clear.

I would prefer (personally me) Accession numbers at NCBI home page, but is a matter of taste.